# MMed-RAG: Versatile Multimodal RAG System for Medical Vision Language Models

**Peng Xia**[1], **Kangyu Zhu**[5], **Haoran Li**[6], **Tianze Wang**[3], **Weijia Shi**[4],
**Sheng Wang**[4], **Linjun Zhang**[3], **James Zou**[2], **Huaxiu Yao**[1]
[1]UNC-Chapel Hill, [2]Stanford University, [3]Rutgers University,
[4]University of Washington, [5]Brown University, [6]PloyU
{pxia,huaxiu}@cs.unc.edu

## Abstract

Artificial Intelligence (AI) has demonstrated significant potential in healthcare, particularly in disease diagnosis and treatment planning. Recent progress in Medical Large Vision-Language Models (Med-LVLMs) has opened up new possibilities for interactive diagnostic tools. However, these models often suffer from factual hallucination, which can lead to incorrect diagnoses. Fine-tuning and retrieval-augmented generation (RAG) have emerged as methods to address these issues. However, the amount of high-quality data and distribution shifts between training data and deployment data limit the application of fine-tuning methods. Although RAG is lightweight and effective, existing RAG-based approaches are not sufficiently general to different medical domains and can potentially cause misalignment issues, both between modalities and between the model and the ground truth. In this paper, we propose a versatile multimodal RAG system, MMed-RAG, designed to enhance the factuality of Med-LVLMs. Our approach introduces a domain-aware retrieval mechanism, an adaptive retrieved contexts selection, and a provable RAG-based preference fine-tuning strategy. These innovations make the RAG process sufficiently general and reliable, significantly improving alignment when introducing retrieved contexts. Experimental results across five medical datasets (involving radiology, ophthalmology, pathology) on medical VQA and report generation demonstrate that MMed-RAG can achieve an average improvement of 43.8% in the factual accuracy of Med-LVLMs.

## 1 Introduction

Artificial Intelligence (AI) has already transformed healthcare and still has a lot of potential for further advancements (Tăuţan et al., 2021; Wang et al., 2019; Ye et al., 2021; Tu et al., 2024). Recently, Medical Large Vision-Language Models (Med-LVLMs) have shown great promise for advancing interactive and intelligent diagnosis (Li et al., 2023a; Moor et al., 2023; Zhang et al., 2023b; Wu et al., 2023b). Despite this potential (Li et al., 2023b; Wu et al., 2023a; Shi et al., 2024), current Med-LVLMs still face significant reliability issues, particularly their tendency to generate non-factual medical responses (Xia et al., 2024a; Royer et al., 2024; Chen et al., 2024a; Jiang et al., 2024), making them unreliable in critical medical applications. These factuality issues raise serious concerns when deploying such models in clinical settings, where even small diagnostic errors could lead to severe consequences for patient care.

Recently, researchers have begun to focus on improving the factuality of Med-LVLMs through various techniques, including fine-tuning (Li et al., 2023a; Moor et al., 2023; Thawkar et al., 2023; Zhang et al., 2023b; Chen et al., 2024b) and retrieval-augmented generation (RAG) (Xia et al., 2024b; He et al., 2024; Sun et al., 2024b). Fine-tuning is a direct method to improve model performance, but it faces several limitations in the medical field. First, there is a lack of sufficient high-quality labeled data for fine-tuning in the medical domain. Additionally, a distribution gap often exists between the training data and the real-world deployment data (Schrouff et al., 2022), leading to significantly worse model performance during deployment. Hence, RAG has emerged as a viable alternative by providing external references during the inference stage, enhancing the factuality of Med-LVLMs (Wu et al., 2023c; Gao et al., 2023). However, despite its advantages, current RAG implementations in Med-LVLMs have significant limitations. First, these methods tend to

be *dataset-specific*, reducing their generalizability across various medical domains. Second, these models are still facing *misalignment issues* that lead to factuality problems. This misalignment may arise from the impact of adding RAG on the original Med-LVLMs' *cross-modality alignment*, as well as on the *overall alignment* between the model and ground truth.

To address these challenges, we propose a versatile factual **M**ultimodal **Med**ical **RAG** system called **MMed-RAG**. Specifically, MMed-RAG first introduces a domain-aware retrieval mechanism, designed to handle different domains of medical images more effectively. Here, we design a domain identification module to adaptively select a corresponding retrieval model given the input medical image. Secondly, we include a adaptive calibration approach for selecting the number of retrieved contexts. Lastly, MMed-RAG incorporates RAG-based preference fine-tuning to enhance cross-modality alignment and overall alignment with ground truth. The preference pairs are designed to achieve two goals: first, to improve cross-modality alignment by encouraging the model to avoid generating responses without utilizing input medical images, even the responses are correct; second, to improve overall alignment by encouraging the model to understand retrieved contexts when unsure, while avoiding interference from irrelevant retrieved information.

The primary contribution of this paper is MMed-RAG, a versatile multimodal RAG system designed specifically for Med-LVLMs to generate more factual responses. Under mild assumptions, our theoretical analysis demonstrates that MMed-RAG mitigates both cross-modality misalignment and overall misalignment with ground truth. Furthermore, empirical results on five medical multimodal datasets, covering three medical image modalities (radiology, pathology, and ophthalmology), show that MMed-RAG significantly improves the factual accuracy of Med-LVLMs, achieving improvements of 18.5% and 69.1% on Medical VQA and report generation tasks, respectively, compared to the original Med-LVLM. These empirical findings further demonstrate the effectiveness of our proposed components and support the theoretical analysis in addressing misalignment issues.

## 2    Preliminaries

In this section, we will provide a brief overview of Med-LVLMs and preference optimization.

**Medical Large Vision Language Models**. Med-LVLMs bridge LLMs with medical visual modules, allowing the model to take medical image $x_v$ and clinical query $x_t$ as input $x$, and autoregressively predict the probability distribution of the next token. The text output is denoted as $y$.

**Preference Optimization**. Preference optimization has achieved remarkable results in LLM alignment. Give an input $x$, a language model policy $\pi_\theta$ can produce a conditional distribution $\pi_\theta(y \mid x)$ with $y$ as the output text response. The recently popular DPO (Rafailov et al., 2023) utilizes preference data achieve objective alignment in LLMs. The preference data is defined as $\mathcal{D} = \{x^{(i)}, y_w^{(i)}, y_l^{(i)}\}_{i=1}^N$, where $y_w^{(i)}$ and $y_l^{(i)}$ represent preferred and dispreferred responses given an input prompt $x$. The probably of obtaining each preference pair is $p(y_w \succ y_l) = \sigma(r(x, y_w) - r(x, y_l))$, where $\sigma(\cdot)$ is the sigmoid function. In DPO, the optimization can be formulated as classification loss over the preference data as:

$$\mathcal{L}_{DPO}(\pi_\theta; \pi_{\text{ref}}) = -\mathbb{E}_{(x, y_w, y_l) \sim \mathcal{D}} \left[ \log \sigma \left( \alpha \log \frac{\pi_\theta(y_w|x)}{\pi_{\text{ref}}(y_w|x)} - \alpha \log \frac{\pi_\theta(y_l|x)}{\pi_{\text{ref}}(y_l|x)} \right) \right]. \tag{1}$$

where $\pi_\theta$ represents the reference policy, which is the LLM fine-tuned through supervised learning.

## 3    MMed-RAG: A Versatile Medical RAG System

In this section, as illustrated in Figure 1, we will propose MMed-RAG, a versatile RAG system for improving the factuality of Med-LVLMs. Specifically, MMed-RAG consists of three complementary modules. First, we design a domain-aware retrieval mechanism to select the optimal retriever by feeding each given medical image to the domain identification module. Second, to select an optimal number of retrieved contexts and filter out low-quality information, MMed-RAG adopts a adaptive method by filtering out low-quality information using the similarity scores during the RAG phase. Lastly, we use a RAG-based preference fine-tuning approach to improve the cross-modality alignment and the overall alignment between groundtruth. We detail these steps as follows:

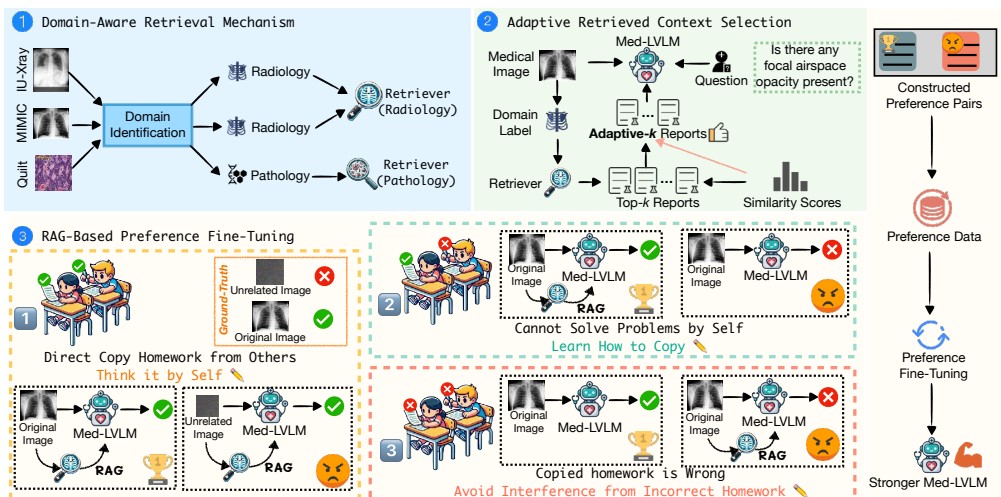

Figure 1: Overview of MMed-RAG, a versatile factual multimodal RAG system designed to enhance the reliability of Med-LVLMs. It introduces a domain-aware retrieval mechanism that effectively handles different domains of medical images by selecting suitable retrieval models. Additionally, it uses an adaptive context selection approach to determine the optimal number of retrieved contexts and employs preference fine-tuning to improve both cross-modality and overall alignment.

## 3.1 DOMAIN-AWARE RETRIEVAL MECHANISM

In MMed-RAG, we introduce a domain-aware retrieval mechanism to efficiently handle medical images from different sources (e.g., radiology, pathology, ophthalmology). Specifically, we first employ a domain identification module that assigns a domain label to each input medical image. To achieve this, we create a small dataset with medical images as inputs and their corresponding domain labels as outputs, using this dataset to fine-tune the BiomedCLIP model (Zhang et al., 2023a) to improve its domain awareness. Formally, for a given medical image $x_v$, we predict its domain $d = \mathcal{F}(x_v)$. Based on the assigned domain label $d$, the image $x_v$ is fed into the corresponding multimodal retriever $\mathcal{R}_d(\cdot)$ for knowledge retrieval.

Here, each multimodal retriever $\mathcal{R}_d(\cdot)$ for each domain $d$ is trained through contrastive learning (Radford et al., 2021). Specifically, the visual and textual information $X_{img}, X_{txt}$ are processed by their corresponding encoders $\mathcal{E}_{img}(\cdot), \mathcal{E}_{txt}(\cdot)$ to generate textual and visual embeddings $V_{txt} = \mathcal{E}_{txt}(X_{txt}), V_{img} = \mathcal{E}_{img}(X_{img})$. Contrastive learning loss is then applied to maximize the similarity between text and image embeddings representing the same example, while minimizing the similarity between embeddings representing different examples, as defined below:

$$\mathcal{L} = \frac{\mathcal{L}_{img} + \mathcal{L}_{txt}}{2}, \text{where } \mathcal{L}_{img} = -\frac{1}{N}\sum_{i=1}^{N}\log\frac{\exp(S_{i,i})}{\sum_{j=1}^{N}\exp(S_{i,j})}, \mathcal{L}_{txt} = -\frac{1}{N}\sum_{i=1}^{N}\log\frac{\exp(S_{i,i})}{\sum_{j=1}^{N}\exp(S_{j,i})},$$
(2)

where $S \in \mathbb{R}^{N \times N}$ represents the similarity matrix between image and text modalities, calculated as: $S = \frac{V_{img}}{|V_{img}|} \cdot (\frac{V_{txt}}{|V_{txt}|})^T$, where each element $S_{i,j}$ represents the similarity between the image representation of example $i$ and the text representation of example $j$.

Finally, for the input image $x_t$, after feeding into the corresponding multimodal retriever $\mathcal{R}_d(\cdot)$, the multimodal retriever will retrieves the top-$k$ most similar reports for the image. These retrieved reports $x_r = \mathcal{R}_d(x_v)$ are then provided to the Med-LVLM $\mathcal{M}(\cdot)$ as references to guide the generation.

Figure 2: Relations between selected contexts and similarity score.

## 3.2 ADAPTIVE RETRIEVED CONTEXT SELECTION

Following the domain-aware retrieval mechanism, the next step is to determine the optimal amount of context to retrieve.

Retrieving too much or too little information can result in hallucinations (Xia et al., 2024b). Current RAG methods applied to Med-LVLMs generally rely on empirical results or fixed values based on validation sets to select the optimal value of the number of retrieved contexts $k$ (Xia et al., 2024b; He et al., 2024; Sun et al., 2024b). However, the distribution of similarity scores varies depending on the complexity of the image and its alignment with the textual information from the data source. These fixed-$k$ methods do not guarantee optimal performance on target data, as they overlook the similarity scores generated during the retrieval process. To address this, we propose an adaptive method that dynamically selects $k$ based on the similarity scores of the retrieved contexts. Specifically, during the domain-aware retrieval mechanism phase, the retrieved information is denoted as $x_r(k) = \mathcal{R}_d(x_v; k)$, where $k$ represents the number of retrieved contexts, and the corresponding similarity scores are denoted as $S_k$. For simplicity, when there is no ambiguity, we will refer to $x_r(k)$ as $x_r$.

As illustrated in Figure 2, our method is based on a key observation: the similarity scores (CLIP score in this case) between retrieved contexts often exhibit a sharp decline after a certain number of results (nearly top-9 in this case). This suggests that lower-quality information can still be included among the top-$k$ retrieved contexts when using a fixed-$k$ strategy, especially in cases where the fixed value of $k$ is too large. These lower-quality retrievals introduce noise and irrelevant information, which can significantly impair the model's ability to generate factual and coherent responses. To mitigate this issue, we draw inspiration from the Gap statistic method used in clustering (Tibshirani et al., 2001) and extend this concept to RAG for Med-LVLMs. Specifically, after retrieving the top-$k$ contexts, we perform an additional round of $k$ optimization by analyzing the similarity ratios between consecutive retrievals. These similarity ratios are denoted as $u_i = \log(S_i/S_{i+1})$ for $0 < i \le k$, where $S_i$ represents the similarity score of the $i$-th retrieved context. When $u_i$ exceeds a predefined threshold $\gamma$, this indicates a substantial drop in relevance, suggesting that the remaining retrievals are less likely to contribute preferredly to the model's output. At this point $i$, we truncate $k$, effectively discarding the less relevant retrievals that follow. This adaptive truncation mechanism ensures that only the most relevant contexts are retained for generating the final response, reducing the risk of hallucination and improving the factual accuracy of the outputs.

Although the threshold $\gamma$ is fixed, this approach provides a adaptive way to balance the bias and variance in retrieved contexts. By adapting to the characteristics of each input $x_v$, our method enhances the robustness of the retrieval process and ensures that the selection of $k$ is tailored to the specific data at hand, thereby improving overall performance across diverse contexts and tasks.

### 3.3 RAG-BASED PREFERENCE FINE-TUNING

After context selection, MMed-RAG supplies Med-LVLM with reliable retrieved information as external knowledge to aid in generating factual responses. However, incorporating this retrieved knowledge may potentially disrupt the original alignment within the existing Med-LVLM, a concern we will elaborate on below:

**Alignment Analysis.** In the alignment analysis, we aim to explore how incorporating retrieved context impacts the original alignment in Med-LVLMs, focusing on two key aspects: (1) cross-modality alignment and (2) overall alignment with the ground truth. To evaluate cross-modality alignment, we conduct two tests on LLaVA-Med-1.5 (Li et al., 2023a) using the Harvard-FairVLMed (Luo et al., 2024) dataset. First, when replacing the original image with a highly noisy image associated with a different ground truth, the original model gives incorrect answers (the ground truth being the response for the original image). After incorporating RAG, where context is retrieved based on the original image, 55.08% of these cases return correct answers. This indicates that the model *directly references the retrieved knowledge* without considering the input image, highlighting significant *cross-modal misalignment issues*. Furthermore, 43.31% of the questions that were originally answered correctly are answered incorrectly after incorporating RAG, suggesting *interference from incorrect retrieval information*, which leads to *overall misalignment with the ground truth*.

To address cross-modality misalignment and the overall misalignment introduced by incorporating retrieved knowledge, as shown in Algorithm 1, we propose a RAG-based preference fine-tuning (RAG-PT) approach to fine-tune the target Med-LVLM $\mathcal{M}(\cdot)$. Specifically, RAG-PT constructs two types of preference pairs designed to mitigate both categories of misalignment.

**Preference Pairs for Cross-Modality Alignment.** We first construct preference pairs aimed at improving cross-modality alignment. In this dataset, we select samples from $\mathcal{D} = \{x_v^{(i)}, x_t^{(i)}, y^{(i)}\}_{i=1}^N$,

---

**Algorithm 1:** Versatile Multimodal RAG System (MMed-RAG)

---

**Input:** $\mathcal{D} = \{x_v^{(i)}, x_t^{(i)}, y^{(i)}\}_{i=1}^N$: Dataset; $\pi_\theta$: Parameters of the Med-LVLM; Med-LVLM: $\mathcal{M}(\cdot, \cdot)$;
Domain Identification: $\mathcal{F}(\cdot)$; Retriever: $\mathcal{R}(\cdot)$; Noisy Function: $\mathcal{I}(\cdot)$.
**Output:** $\pi_{\text{ref}}$: Parameters of the reference model.

1 ▷ *Training Stage*
2 Initialize $\mathcal{D}_{cm}$ with an empty set
3 **foreach** $\underline{(x_v, x_t, y) \in \mathcal{D}}$ **do**
4      Generate retrieved contexts with an assigned domain label $x_r \leftarrow \mathcal{R}_{\mathcal{F}(x_v)}(x_v)$
5      Generate the noisy image $x_v^* \leftarrow \mathcal{I}(x_v)$
6      ▷ *Cross-Modality Alignment*
7      **if** $\underline{\mathcal{M}(x_v, (x_t, x_r)) = y \text{ and } \mathcal{M}(x_v^*, (x_t, x_r)) = y}$ **then**
8          Select the preferred response $y_{w,o1} \leftarrow y$, dispreferred response $y_{l,o1} \leftarrow \mathcal{M}(x_v^*, (x_t, x_r))$
9          Put $\{(x_v, x_t), y_{w,o1}, y_{l,o1}\}$ into $\mathcal{D}_{cm}$
10      ▷ *Overall Alignment*
11      Initialize $\mathcal{D}_{oa}^1$ and $\mathcal{D}_{oa}^2$ with empty set
12      **if** $\underline{\mathcal{M}(x_v, (x_t, x_r)) = y \text{ and } \mathcal{M}(x_v, x_t) \neq y}$ **then**
13          Select the preferred response $y_{w,o2} \leftarrow y$, dispreferred response $y_{l,o2} \leftarrow \mathcal{M}(x_v, x_t)$
14          Put $\{(x_v, x_t), y_{w,o2}, y_{l,o2}\}$ into $\mathcal{D}_{oa}^1$
15      **if** $\underline{\mathcal{M}(x_v, x_t) = y \text{ and } \mathcal{M}(x_v, (x_t, x_r)) \neq y}$ **then**
16          Select the preferred response $y_{w,o3} \leftarrow y$, dispreferred response $y_{l,o3} \leftarrow \mathcal{M}(x_v, (x_t, x_r))$
17          Put $\{(x_v, x_t), y_{w,o3}, y_{l,o3}\}$ into $\mathcal{D}_{oa}^2$
18 $\mathcal{D}_{pt} = \mathcal{D}_{cm} \cup \mathcal{D}_{oa}, \mathcal{D}_{oa} = \mathcal{D}_{oa}^1 \cup \mathcal{D}_{oa}^2$
19 **foreach** $\underline{((x_v, x_t), y_{w,o}, y_{l,o}) \in \mathcal{D}_{pt}}$ **do**
20      Compute the losses $\mathcal{L}_{pt}$ following equation 4 and update $\pi_{\text{ref}}$
21 ▷ *Inference Stage*
22 **foreach** $\underline{\text{test sample } (x_v, x_t)}$ **do**
23      Select top-k retrieved contexts with an assigned domain label $x_r \leftarrow \mathcal{R}_{\mathcal{F}(x_v)}(x_v)$
24      Get the predictions of the model w/ RAG-PT $p \leftarrow \mathcal{M}(x_v, (x_t, x_r))$

---

where $x_v$, $x_t$, and $y$ represent the input medical image, clinical query, and ground-truth answer, respectively. For simplicity, we omit the sample index $(i)$ in the following sections. A model's correct response using retrieved knowledge, i.e., $\mathcal{M}(x_v, x_t + x_r) = y$, is considered a preferred response $p_i$, where $x_r$ is the retrieved information. A dispreferred response $n_i$ is selected from cases where the model makes a correct inference based on an unrelated image, i.e., $\mathcal{M}(x_v^*, x_t) \neq y$, but $\mathcal{M}(x_v^*, x_t + x_r) = y$, reflecting the model's reliance on the retrieved knowledge. The unrelated images $x_v^*$ are generated through a two-step process: first, we use the retriever to select an image $x_v'$ with the lowest similarity to the target image; then, we introduce diffusion noise into the selected unrelated image. We define the noise step as $s$, and the noised image at step $s$ is expressed as:

$$x_v^* = \sqrt{\bar{\xi}_s} \cdot x_v' + \sqrt{1 - \bar{\xi}_s} \cdot \epsilon, \tag{3}$$

where $\bar{\xi}_s = \prod_{i=0}^s \xi_i$ and $\xi_s \in (0, 1)$ is a hyperparameter. The preference pairs constructed in this stage are denoted as $\mathcal{D}_{cm}$. By comparing the preferred and dispreferred responses in $\mathcal{D}_{cm}$, we encourage the model to prioritize the input medical image when generating responses.

**Preference Pairs for Overall Alignment.** Second, we construct preference pairs to improve overall alignment, focusing on enhancing the model's ability to effectively leverage retrieved knowledge when generating responses. The preference pairs in this stage are constructed from two subsets. The first subset, $\mathcal{D}_{oa}^1$, is designed to strengthen the model's comprehension and reasoning abilities regarding the retrieved knowledge. Preferred responses are selected where the model correctly answers based on both the original image and the retrieved information, i.e., $\mathcal{M}(x_v, x_t + x_r) = y$, while dispreferred responses represent cases where the model answers incorrectly based on the image without using retrieval, i.e., $\mathcal{M}(x_v, x_t) \neq y$. Comparing these preferred and dispreferred responses enhances the model's understanding of the retrieved information and improves the overall effectiveness of RAG. In the second subset, $\mathcal{D}_{oa}^2$, the goal is to mitigate interference from the retrieved knowledge. Preferred responses are selected where the model correctly answers based solely on the original image without using retrieved knowledge, i.e., $\mathcal{M}(x_v, x_t) = y$, while dispreferred responses occur when the model answers incorrectly using both the image and retrieved information, i.e., $\mathcal{M}(x_v, x_t + x_r) \neq y$. This helps the model learn when to rely on its internal knowledge

versus retrieved knowledge. Finally, we combine the first and second subsets to form the second set of preference pairs, $\mathcal{D}_{oa} = \mathcal{D}_{oa}^1 \cup \mathcal{D}_{oa}^2$.

Finally, we merge the first and second preference set and denote the preference dataset as $\mathcal{D}_{pt} = \mathcal{D}_{cm} \cup \mathcal{D}_{oa} = \{x^{(i)}, y_{w,o}^{(i)}, y_{l,o}^{(i)}\}_{i=1}^N$, where $y_{w,o}^{(i)}$, $y_{l,o}^{(i)}$ are represented as preferred and dispreferred responses, respectively. Based on the curated preferences, we fine-tune Med-LVLM using direct preference optimization (Rafailov et al., 2023) with the following loss:

$$\mathcal{L}_{pt} = -\mathbb{E}_{(x,y_{w,o},y_{l,o})\sim\mathcal{D}} \left[ \log \sigma \left( \alpha \log \frac{\pi_\theta(y_{w,o}|x)}{\pi_o(y_{w,o}|x)} - \alpha \log \frac{\pi_\theta(y_{l,o}|x)}{\pi_o(y_{l,o}|x)} \right) \right]. \tag{4}$$

## 4 EXPERIMENT

In this section, we evaluate the performance of MMed-RAG, aiming to answer the following questions: (1) Can MMed-RAG effectively improve the factuality of Med-LVLMs compared to decoding-based and RAG-based baselines? (2) How effective is each proposed component on performance? (3) What is the effect of preference data for different alignment goals? and (4) Does MMed-RAG actually improve cross-modality alignment and overall alignment?

### 4.1 EXPERIMENTAL SETUPS

**Implementation Details**. We use LLaVA-Med-1.5 7B (Li et al., 2023a) as the backbone model. During the preference fine-tuning process, we adapt LoRA fine-tuning (Hu et al., 2021). For the training of retriever, the vision encoder is a ResNet-50 (He et al., 2016), and the text encoder is a bio-BioClinicalBERT (Alsentzer et al., 2019). We use the AdamW optimizer with a learning rate of $10^{-3}$, weight decay of $10^{-2}$ and a batch size of 32. The model is trained for 360 epochs. For more detailed information on training hyperparameters and training data, please see Appendix A.1.1.

**Baseline Methods**. We compare MMed-RAG with two types of LVLM hallucination mitigation methods that show promising results in natural image understanding. 1) Decoding-based methods, including Greedy Decoding, Beam Search (Sutskever et al., 2014), DoLa (Chuang et al., 2023), OPERA (Huang et al., 2023), VCD (Leng et al., 2023). These methods manipulate the logits of the model's output tokens to enhance factual accuracy. 2) Multimodal RAG-based methods, including MedDr (He et al., 2024), FactMM-RAG (Sun et al., 2024b), RULE (Xia et al., 2024b). Furthermore, we compare the performance with other open-source Med-LVLMs, including Med-Flamingo (Moor et al., 2023), MedVInT (Zhang et al., 2023b), RadFM (Wu et al., 2023b).

**Evaluation Datasets**. We utilize five medical vision-language datasets for medical VQA and report generation tasks, i.e., MIMIC-CXR (Johnson et al., 2019), IU-Xray (Demner-Fushman et al., 2016), Harvard-FairVLMed (Luo et al., 2024), PMC-OA (Lin et al., 2023a) (we only select the pathology part) and Quilt-1M (Ikezogwo et al., 2024). These datasets cover radiology, ophthalmology, and pathology. To construct the VQA benchmarks, following (Xia et al., 2024a), we generate question-answer pairs from medical reports using GPT-4 (OpenAI, 2023), with answers formatted as *yes* or *no*. Pathology images are excluded from the report generation task due to their brief and insufficient descriptions. The detailed dataset descriptions are provided in the Appendix A.2.

**Evaluation Metrics**. Following (Jing et al., 2017; Lin et al., 2023b), we use Accuracy, F1 Score and AUROC for evaluating medical VQA task, and BLEU Score (Papineni et al., 2002), ROUGE-L (Lin, 2004) and METEOR (Banerjee & Lavie, 2005) for evaluating report generation task.

### 4.2 MAIN RESULTS

In this section, we provide a comprehensive comparison with various baseline methods and other open-source Med-LVLMs on medical VQA and report generation tasks.

**Comparison with Baselines.** We compare MMed-RAG with baseline methods on medical VQA and report generation tasks, with the results presented in Table 1 and Table 2, respectively. Overall, MMed-RAG outperforms all baselines across nearly all metrics and datasets. Specifically, MMed-RAG demonstrates a significant performance boost, improving by 18.5% and 69.1% over the original Med-LVLM in medical VQA and report generation tasks, respectively. When compared to baseline methods, MMed-RAG surpasses decoding-based approaches, achieving improvements of

Table 1: Model performance (%) of different methods based on LLaVA-Med-1.5 on medical VQA task. Notably, we report the accuracy, F1 score and AUROC. The best results and second best results are highlighted in red and blue, respectively.

| Models | Radiology | | | | | | Ophthalmology | | | Pathology | | | | | |
| --- | --- | --- | --- | --- | --- | --- | --- | --- | --- | --- | --- | --- | --- | --- | --- |
| | IU-Xray | | | MIMIC-CXR | | | Harvard-FairVLMed | | | Quilt-1M | | | PMC-OA (Pathology) | | |
| | Acc | F1 | AUC | Acc | F1 | AUC | Acc | F1 | AUC | Acc | F1 | AUC | Acc | F1 | AUC |
| LLaVA-Med-1.5 | 75.47 | 64.04 | 67.46 | 75.79 | 80.49 | 68.84 | 63.03 | 74.11 | 63.05 | 62.80 | 72.90 | 60.03 | 59.28 | 71.98 | 54.19 |
| + Greedy | 76.88 | 65.59 | 68.74 | 78.32 | 86.75 | 71.13 | 82.54 | 85.98 | 70.09 | 64.72 | 70.12 | 58.75 | 58.61 | 70.42 | 53.10 |
| + Beam Search | 76.91 | 66.06 | 68.77 | 81.56 | 86.36 | 73.79 | 80.93 | 88.08 | 68.94 | 63.52 | 69.33 | 57.65 | 56.29 | 69.84 | 52.89 |
| + DoLa | 78.00 | 66.75 | 72.19 | 81.35 | 85.73 | 72.73 | 76.87 | 85.53 | 67.10 | 63.47 | 69.10 | 57.58 | 57.71 | 70.27 | 52.95 |
| + OPERA | 70.59 | 61.54 | 63.22 | 69.34 | 76.66 | 62.46 | 71.41 | 81.37 | 65.59 | 60.51 | 66.32 | 54.79 | 55.32 | 68.30 | 51.86 |
| + VCD | 68.99 | 54.35 | 61.08 | 70.89 | 75.57 | 64.61 | 65.88 | 77.20 | 64.16 | 61.43 | 67.39 | 55.72 | 55.10 | 67.94 | 51.62 |
| + MedDr | 83.33 | 67.80 | 77.15 | 55.16 | 56.18 | 58.47 | 70.17 | 80.72 | 64.15 | 68.15 | 73.23 | 67.01 | 59.97 | 69.19 | 57.01 |
| + FactMM-RAG | 84.51 | 68.51 | 77.07 | 77.58 | 81.86 | 70.09 | 83.67 | 87.21 | 72.20 | 69.25 | 73.62 | 68.15 | 60.49 | 69.38 | 57.31 |
| + RULE | 87.84 | 78.00 | 85.78 | 83.92 | 87.49 | 83.44 | 87.12 | 92.89 | 77.08 | 68.97 | 73.80 | 68.13 | 61.41 | 70.36 | 58.91 |
| MMed-RAG | 89.54 | 80.72 | 87.13 | 83.57 | 88.49 | 85.08 | 87.94 | 92.78 | 80.81 | 72.95 | 76.35 | 72.25 | 64.54 | 73.09 | 61.42 |

Table 2: Model performance (%) of different methods based on LLaVA-Med-1.5 on report generation task. Notably, we report the average BLEU, ROUGE-L, METEOR.

| Models | Radiology | | | | | | Ophthalmology | | |
| --- | --- | --- | --- | --- | --- | --- | --- | --- | --- |
| | IU-Xray | | | MIMIC-CXR | | | Harvard-FairVLMed | | |
| | BLEU | ROUGE-L | METEOR | BLEU | ROUGE-L | METEOR | BLEU | ROUGE-L | METEOR |
| LLaVA-Med-1.5 | 9.64 | 12.26 | 8.21 | 12.11 | 13.05 | 11.16 | 18.11 | 11.36 | 10.75 |
| + Greedy | 11.47 | 15.38 | 12.69 | 16.63 | 14.26 | 14.19 | 17.98 | 11.49 | 13.77 |
| + Beam Search | 12.10 | 16.21 | 13.17 | 16.97 | 14.74 | 14.43 | 18.37 | 12.62 | 14.50 |
| + DoLa | 11.79 | 15.82 | 12.72 | 17.11 | 14.89 | 14.81 | 18.26 | 12.51 | 14.51 |
| + OPERA | 10.66 | 14.70 | 12.01 | 15.40 | 12.52 | 13.72 | 16.59 | 11.47 | 13.63 |
| + VCD | 10.42 | 14.14 | 11.59 | 15.18 | 12.30 | 13.38 | 16.73 | 11.38 | 13.89 |
| + MedDr | 12.37 | 16.45 | 13.50 | 18.59 | 15.72 | 16.77 | 19.82 | 13.72 | 15.40 |
| + FactMM-RAG | 14.70 | 18.05 | 15.92 | 18.71 | 15.84 | 16.82 | 20.82 | 14.17 | 15.31 |
| + RULE | 27.53 | 23.16 | 27.99 | 18.61 | 15.96 | 17.42 | 22.35 | 14.93 | 17.74 |
| MMed-RAG | 31.38 | 25.59 | 32.43 | 23.25 | 12.34 | 20.47 | 24.82 | 16.59 | 19.85 |

11.5% and 44.2% in the two tasks. Furthermore, recent RAG-based methods show substantial improvements over earlier techniques, yet our approach still outperforms RAG-based baselines by 2.8% and 16.1% in the medical VQA and report generation tasks, respectively. This indicates that MMed-RAG effectively mitigates misalignment issues introduced by RAG. Notably, MMed-RAG achieves more pronounced gains in report generation, likely due to the higher complexity of the task and the greater influence of retrieved contexts in guiding open-ended generation.

**Comparison with Other Med-LVLMs.** To provide a comprehensive comparison, we evaluate MMed-RAG against other open-source Med-LVLMs to demonstrate the superiority of our approach. We assess the performance of these models across different medical image modalities, reporting the average results for medical VQA and report generation tasks in Table 3 (see Appendix A.6 for detailed results). Our findings show that MMed-RAG significantly outperforms Med-LVLMs pre-trained on large-scale datasets across various domains. This reinforces the generalizability and effectiveness of our approach across diverse image domains and medical multimodal tasks.

Table 3: Performance comparison with several Med-LVLMs. Rad: Radiology, Opt: Ophthalomology, Pat: Pathology.

| Model | Rad | Opt | Pat |
| --- | --- | --- | --- |
| Med-Flamingo | 27.42 | 22.50 | 29.11 |
| MedVInT | 33.17 | 29.40 | 25.33 |
| RadFM | 35.82 | 27.07 | 24.82 |
| miniGPT-Med | 36.66 | 25.28 | 23.16 |
| MMed-RAG | **56.94** | **56.38** | **54.10** |

## 4.3 ANALYSIS

In this section, we provide a detailed analysis of each module's performance, along with a series of analytical experiments, to better understand the performance gains of MMed-RAG. Additionally, we demonstrate the compatibility of our approach, achieving a consistent 40.3% performance improvement on a different backbone, i.e., LLAVA-Med-1.0 (see details in Appendix A.6).

Table 4: Ablation results on two datasets covering different domains. RG: report generation, FairVLMed: Harvard-FairVLMed.

| Model | IU-Xray | | FairVLMed | |
| --- | --- | --- | --- | --- |
| | VQA | RG | VQA | RG |
| LLaVA-Med-1.5 | 68.99 | 10.04 | 66.63 | 13.41 |
| +DR | 77.12 | 13.23 | 72.69 | 15.89 |
| +RCS | 79.56 | 17.92 | 75.74 | 17.22 |
| +RAG-PT (Ours) | **85.80** | **29.80** | **87.18** | **20.42** |

**Ablation Studies.** We conduct a series of ablation experiments to evaluate the impact of each component in MMed-RAG. The results for both medical VQA and report generation tasks on the IU-Xray and Harvard-FairVLMed datasets are summarized in Table 4. According to the results, we can see that: (1) The domain-aware retrieval mechanism (DR) significantly improves the factuality of Med-LVLM, with an average performance increase of 17.9% and 16.1% on the IU-Xray and FairVLMed datasets, respectively. Here, the retrieved knowledge aids the model in generating more factual responses. (2) Building on this, the introduction of adaptive retrieval context selection (RCS) further filters out unreliable retrieved contexts, yielding an additional performance boost of 19.3% and 6.3% on the IU-Xray and FairVLMed datasets. (3) The inclusion of RAG-based preference fine-tuning (RAG-PT) enhances the model's understanding of the retrieved knowledge, leading to substantial performance gains of 37.1% and 16.9% on the respective datasets. This demonstrates that RAG-PT effectively addresses misalignment issues.

**Impact of the Preference Data in RAG-PT.** To better understand how RAG-PT mitigates the misalignment issue and improves performance, we conducted a detailed study on the training preference data composition of RAG-PT. As described in Section 3.3, the RAG-PT data is designed to address both cross-modality alignment and overall alignment objectives, with the latter focusing on enhanced understanding of retrieved knowledge and minimizing retrieval interference.

Table 5: Performance using RAG-PT based on subsets of preference data.

| Model | IU-Xray | | FairVLMed | |
|---|---|---|---|---|
| | VQA | RG | VQA | RG |
| LLaVA-Med-1.5 | 68.99 | 10.04 | 66.63 | 13.41 |
| +RAG-PT 1 | 80.19 | 19.38 | 79.42 | 18.37 |
| +RAG-PT 2 | 80.27 | **20.16** | 79.35 | 18.66 |
| +RAG-PT 3 | **81.30** | 19.43 | **80.07** | **18.92** |

The detailed experimental results in Table 5 demonstrate that the preference data tailored for different alignment objectives positively impacts the model's performance, showing the effectiveness of RAG-PT.

**How Effective is MMed-RAG in Mitigating Misalignment Issues?** To gain a more intuitive understanding of the effectiveness of MMed-RAG in addressing misalignment issues: 1) we calculate the proportion of errors caused by RAG and compare it to the proportion after incorporating MMed-RAG. 2) We visualize the attention maps of image and text tokens with and without RAG-PT. First, as mentioned in Section 3.3, the model may directly copy reference information, referred to as Copy-Reference (CR) rate. After applying MMed-RAG, as shown in Figure 3, the CR rate drops to 28.19%. Additionally, the proportion of errors affected by RAG interference, referred to as Over-Reliance (OR) rate, which is initially 43.31%, decreased to 8.38% after incorporating MMed-RAG. Furthermore, as shown in Figure 4, the original Med-LVLM tends to rely more heavily on text while ignoring visual information. When retrieval information is introduced, the original Med-LVLM focused more on the retrieved answers, even if the content is incorrect. After RAG-PT, the model significantly increases its attention to visual information and reduces the interference of RAG, thus better aligning the model's knowledge with the fundamental facts.

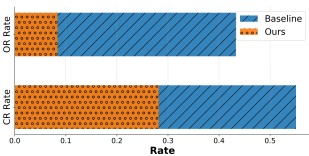

Figure 3: Alignment analysis with and without RAG. OR: Over-Reliance; CR: Copy-Reference.

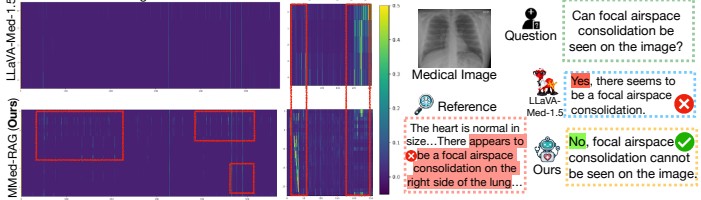

Figure 4: Visualization of attention map. The red box region is labeled with the attentions that can be enhanced by MMed-RAG.

## 5 CONCLUSION

This paper introduces MMed-RAG, a versatile multimodal RAG system designed to address the critical issue of factual hallucination in Med-LVLMs. MMed-RAG employs a domain-aware retrieval mechanism, adaptive calibration for selecting the optimal number of retrieved contexts, and RAG-based preference fine-tuning to improve both cross-modality alignment and overall alignment with the ground truth. These enhancements significantly boost the factual accuracy of Med-LVLMs. Experimental results demonstrate MMed-RAG' effectiveness in enhancing factual accuracy across various imaging domains, underscoring its potential for reliable use in healthcare.

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

# A  EXPERIMENT

## A.1  EXPERIMENTAL SETUP

### A.1.1  DATA STATISTICS

The data quantities used in this study are presented in Table 6, Table 7 and Table 8. We clarify that for training the retriever, the data refers to the number of image-text pairs, while for fine-tuning, it refers to the number of QA items. The "All" category represents the total amount of data used to construct the preference dataset for RAG-PT. The training of RAG-PT includes three types of samples: (a) clean samples with originally correct answers that remain correct even after adding noise to the images, (b) clean image samples with originally incorrect answers that become correct, and (c) clean image samples with originally correct answers that become incorrect.

Table 6: Data statistics for medical VQA task. "Train (DR)" refers to the number of image-text pairs for retriever training, "All (RAG-PT)" refers to the total data for RAG-PT, and "Train (RAG-PT)-a/b/c" refer to the respective subsets for RAG-PT training.

| Dataset | Train (DR) | All (RAG-PT) | Train (RAG-PT)-a | Train (RAG-PT)-b | Train (RAG-PT)-c |
|---|---|---|---|---|---|
| Ophthalomology | 7000 | 3247 | 1082 | 1030 | 1135 |
| Radiology | 4034 | 4836 | 1612 | 1989 | 1235 |
| Pathology | 5000 | 1990 | 663 | 523 | 804 |

### A.1.2  HYPERPARAMETER SETTINGS

Following the settings of CLIP (Radford et al., 2021), we adopt the same architecture and hyperparameters for the vision and text encoders. The vision encoder is a ResNet-50 (He et al., 2016), and the text encoder is a bio-bert-based model (Alsentzer et al., 2019). We use the AdamW optimizer with a learning rate of $10^{-4}$ and a batch size of 512. The model is trained for 360 epochs. For the first phase, we trained for 3 epochs, and for the second phase, the training was conducted for 1 epoch.

Table 7: Data statistics for report generation. "Train (DR)" refers to the number of image-text pairs for retriever training, "All (RAG-PT)" refers to the total data for RAG-PT, and "Train (RAG-PT)-a/b/c" refer to the respective sample categories for RAG-PT training.

| Dataset | Train (R) | All (RAG-PT) | Train (RAG-PT)-a | Train (RAG-PT)-b | Train (RAG-PT)-c |
|---|---|---|---|---|---|
| Ophthalmology | 7000 | 3247 | 142 | 78 | 207 |
| Radiology | 4034 | 4836 | 233 | 126 | 342 |

Table 8: Data statistics for various datasets. The rows represent the number of images and QA pairs for each dataset.

| | Harvard-FairVLMed | IU-Xray | MIMIC-CXR | PMC-OA | Quilt-1M |
|---|---|---|---|---|---|
| # Images | 713 | 589 | 700 | 530 | 559 |
| # QA Items | 4285 | 2573 | 3470 | 3124 | 1994 |

Training for 20 hours on one A100 80G GPU. For the RAG-PT phase, we adjust the diffusion noise level, symbolized by $\xi$ through a specific formula: $\xi = \text{Sigmoid}(l_t) \times (0.5 \times 10^{-2} - 10^{-5}) + 10^{-5}$, where $\epsilon$ is drawn from a normal distribution. The reports available for retrieval are from the training set of the corresponding dataset. In our experiments, we apply cross-validation to tune all hyper-parameters with grid search. All the experiments are implemented on PyTorch 2.1.2 using four NVIDIA RTX A6000 GPUs. It takes roughly 3 and 4 hours for fine-tuning CLIP and LLaVA-Med-1.5 7B, respectively.

## A.2 EVALUATED DATASETS

We utilize five open-source medical vision-language datasets, i.e., MIMIC-CXR (Johnson et al., 2019), IU-Xray (Demner-Fushman et al., 2016), Harvard-FairVLMed (Luo et al., 2024), PMC-OA (Lin et al., 2023a) and Quilt-1M (Ikezogwo et al., 2024).

- **MIMIC-CXR** (Johnson et al., 2019) is a large publicly available dataset of chest X-ray images in DICOM format with associated radiology reports.

- **IU-Xray** (Demner-Fushman et al., 2016) is a dataset that includes chest X-ray images and corresponding diagnostic reports.

- **Harvard-FairVLMed** (Luo et al., 2024) focuses on fairness in multimodal fundus images, containing image and text data from various sources. It aims to evaluate bias in AI models on this multimodal data comprising different demographics.

- **PMC-OA** (Lin et al., 2023a) is a large-scale dataset comprising figure-caption pairs extracted from PubMed Central. It covers 2,478,267 papers and includes a total of 12,211,907 figure-caption pairs. We only use the pathology subset filtered by GPT-4 based on the captions.

- **Quilt-1M** (Ikezogwo et al., 2024) is the largest vision-language dataset in histopathology, containing 1 million image-text pairs sourced from platforms such as YouTube, Twitter, research papers, and other parts of the internet.

## A.3 EVALUATED MODELS

We evaluate five open-source Med-LVLMs, i.e., LLaVA-Med (Li et al., 2023a), Med-Flamingo (Moor et al., 2023), MedVInT (Zhang et al., 2023b), RadFM (Wu et al., 2023b), miniGPT-Med (Alkhaldi et al., 2024). The selected models are all at the 7B level.

- **LLaVA-Med** (Li et al., 2023a) is a vision-language conversational assistant, adapting the general-domain LLaVA (Liu et al., 2024b) model for the biomedical field. The model is fine-tuned using a novel curriculum learning method, which includes two stages: aligning biomedical vocabulary with figure-caption pairs and mastering open-ended conversational semantics. It demonstrates excellent multimodal conversational capabilities.

- **Med-Flamingo** (Moor et al., 2023) is a multimodal few-shot learner designed for the medical domain. It builds upon the OpenFlamingo, continuing pre-training with medical image-text data

from publications and textbooks. This model aims to facilitate few-shot generative medical visual question answering, enhancing clinical applications by generating relevant responses and rationales from minimal data inputs.

- **RadFM** (Wu et al., 2023b) serve as a versatile generalist model in radiology, distinguished by its capability to adeptly process both 2D and 3D medical scans for a wide array of clinical tasks. It integrates ViT as visual encoder and a perceiver module, alongside the MedLLaMA language model, to generate sophisticated medical insights for a variety of tasks. This design allows RadFM to not just recognize images but also to understand and generate human-like explanations.

- **MedVInT** (Zhang et al., 2023b), which stands for Medical Visual Instruction Tuning, is designed to interpret medical images by answering clinically relevant questions. This model features two variants to align visual and language understanding: MedVInT-TE and MedVInT-TD. Both Med-VInT variants connect a pre-trained vision encoder ResNet-50 adopted from PMC-CLIP (Lin et al., 2023a), which processes visual information from images. It is an advanced model that leverages a novel approach to align visual and language understanding.

- **miniGPT-Med** (Alkhaldi et al., 2024) is a vision-language model derived from large-scale language models and tailored for radiology diagnosis applications. It handles various medical vision-language task using distinct task identifiers, demonstrating advanced performance in disease grounding, medical report generation, and medical VQA.

### A.4 OVERVIEW OF THE BASELINES

We compare MMed-RAG with two types of LVLM hallucination mitigation methods that show promising results in natural image understanding. 1) Decoding-based methods, including Greedy Decoding, Beam Search (Sutskever et al., 2014), DoLa (Chuang et al., 2023), OPERA (Huang et al., 2023), VCD (Leng et al., 2023). These methods manipulate the logits of the model's output tokens to enhance factual accuracy. 2) Multimodal RAG-based methods, including MedDr (He et al., 2024), FactMM-RAG (Sun et al., 2024b), RULE (Xia et al., 2024b).

- **Greedy decoding** involves selecting the most probable next token at each step of generation. While it is efficient and straightforward, it can lead to suboptimal outcomes by getting stuck in repetitive or less creative patterns.

- **Beam search** (Sutskever et al., 2014) expands on greedy decoding by maintaining multiple candidate sequences (or "beams") at each step, allowing for a broader exploration of possible outputs. This approach balances quality and diversity by selecting the top-k sequences based on their probabilities, resulting in more coherent and creative text generation compared to greedy decoding.

- **DoLa** (Chuang et al., 2023) derives the next-token distribution by contrasting the logits projected from later layers against those from earlier layers, leveraging the fact that factual knowledge in LLMs is typically localized within specific transformer layers.

- **OPERA** (Huang et al., 2023) is a LVLMs decoding method based on an Over-trust Penalty and a Retrospection-Allocation strategy The key insight is that hallucinations are closely tied to knowledge aggregation patterns in the self-attention matrix, where MLLMs tend to focus on summary tokens, neglecting image tokens and resulting in content hallucination.

- **VCD** (Leng et al., 2023) is a decoding method that tackles the object hallucination issue in LVLMs. It contrasts output distributions derived from original and distorted visual inputs to calibrate the model's output without the usage of external tools, reducing the the over-reliance on statistical bias and unimodal priors.

- **MedDr** (He et al., 2024) is a healthcare foundation model built upon generated diagnosis-based datasets, demonstrating advanced capabilities in various data modalities. Meddr also integrates a retrieval-augmented medical diagnosis strategy during inferencing to enhance factual accuracy.

- **FactMM-RAG** (Sun et al., 2024b) is a fact-aware multimodal retrieval-augmented pipeline for radiology report generation. It utilize RadGraph to annotate chest radiograph reports and mine clinically relevant pairs to train a universal multimodal retriever.

- **RULE** (Xia et al., 2024b) is an advanced medical retrieval-augmented generation strategy designed to enhance the factuality of Med-LVLMs. First, it introduces a robust strategy for controlling factuality risk through the calibrated selection of retrieved contexts. Second, RULE develops

**Instruction [Round1]**
You are a professional medical expert. I will provide you with some medical reports. Please generate some questions with answers (the answer should be yes or no) based on the provided report. The subject of the questions should be the medical image or patient, not the report.
Below are the given report:
[REPORT]
**Instruction [Round2]**
Please double-check the questions and answers, including how the questions are asked and whether the answers are correct. You should only generate the questions with answers and no other unnecessary information.
Below are the given report and QA pairs in round1:
[REPORT]
[QA PAIRS R1]

Table 9: The instruction to GPT-4 for generating QA pairs.

a preference optimization strategy to balance Med-LVLMs' intrinsic knowledge and the retrieved information.

## A.5 PROMPTS

We convert the medical reports into a series of closed-ended questions with *yes* or *no* answers. To ensure the quality of the VQA data, we perform a round of self-checks using GPT-4 (OpenAI, 2023). Finally, we conduct an round of manual filtering to remove questions with obvious issues or those related to multiple images or patient histories. The prompt templates used are shown in Table 9.

## A.6 ADDITIONAL RESULTS

**Generalization on Different Backbones.** To demonstrate the compatibility of our approach across different backbone models, we apply it to LLaVA-Med-1.0. As shown in Table 10, our method delivers an average improvement of 40.3% over the original LLaVA-Med-1.0, further highlighting its effectiveness in enhancing RAG performance and its adaptability to various backbones. MMed-RAG can be transferred to different Med-LVLMs, yielding consistent improvements across various domains, demonstrating the compatibility of our method.

Table 10: Performance on different backbones.

| Model | IU-Xray | | FairVLMed | |
| --- | --- | --- | --- | --- |
| | VQA | RG | VQA | RG |
| LLaVA-Med-1.0 | 61.73 | 8.74 | 59.54 | 10.59 |
| +MMed-RAG | **80.32** | **22.63** | **78.49** | **15.88** |

**Detailed Results of Other Med-LVLMs.** As shown in Table 11, we illustrate the detailed performance simply shown in Table 3.

## B RELATED WORK

**Factuality in Med-LVLMs.** The rapid advancements in Large Vision-Language Models (LVLMs) (Liu et al., 2024a;b) are beginning to influence the field of medical image analysis. Several Med-LVLMs (Li et al., 2023a; Moor et al., 2023; Zhang et al., 2023b; Wu et al., 2023b), have emerged, showing remarkable performance across different medical imaging modalities. Despite these advances, Med-LVLMs continue to present notable factual hallucination (Xia et al., 2024a; Royer et al., 2024), generating textual outputs that contradict medical visual information. This raises concerns about potential misdiagnoses or overlooked conditions. Recently, benchmarks have been developed to assess the accuracy of Med-LVLMs in tasks such as visual question answering (VQA) and report generation (Xia et al., 2024a; Royer et al., 2024). However, research aimed at enhancing the factual accuracy of Med-LVLMs remains relatively unexplored.

**Retrieval Augmented Generation in Med-LVLMs.** Retrieval-Augmented Generation (RAG) has proven to be a powerful technique for enhancing factual accuracy in language modeling (Gao et al.,

Table 11: Model performance (%) of different Med-LVLMs based on LLaVA-Med-1.5 on medical VQA task.

| Models | Radiology | | Ophthalmology | Pathology | |
|---|---|---|---|---|---|
| | IU-Xray | MIMIC-CXR | Harvard-FairVLMed | Quilt-1M | PMC-OA (Pathology) |
| LLaVA-Med-1.5 | 75.47 | 75.79 | 63.03 | 62.80 | 59.28 |
| MMed-RAG | **89.54** | **83.57** | **87.94** | **72.95** | **64.54** |
| Med-Flamingo | 26.74 | 61.27 | 42.06 | 27.11 | 32.62 |
| MedVInT | 73.34 | 66.06 | 35.92 | 26.81 | 27.77 |
| RadFM | 26.67 | 69.30 | 52.47 | 27.02 | 25.12 |
| miniGPT-Med | 54.87 | 53.92 | 66.73 | 26.82 | 27.03 |

2023; Wu et al., 2023c; Chen et al., 2024c; Qu et al., 2024; Sun et al., 2024a). In the biomedical domain, RAG leverages external knowledge to guide the generation of Med-LVLMs, offering clear advantages in tasks such as medical VQA and report generation (Yuan et al., 2023; Kumar & Marttinen, 2024; Tao et al., 2024; He et al., 2024; Sun et al., 2024b). However, these works mainly focus on enhancing the relevance of the retrieved contexts without considering the model's understanding of retrieved knowledge. Recently, RULE (Xia et al., 2024b) is proposed to use preference fine-tuning to reduce the model's over-reliance on retrieved contexts. However, it still overlooks misalignment issues caused by RAG, as well as the generalizability of the retriever given the diverse domains of input images. In response, we propose MMed-RAG to mitigate these risks, enhancing the factuality of Med-LVLMs by addressing these overlooked factors. This can lead to a better cross-modality and overall alignment to enhance the understanding of retrieved knowledge and visual information, ensuring more consistent and reliable performance across tasks.

## C  THEORETICAL ANALYSIS

In this section, we provide a theoretical analysis of the model obtained from equation 4 and examine how the image input and retrieved context influences the model. Recall that $x_v, y, x_t, x_r$ denotes input medical image, groundtruth answer, question, and retrieved information, respectively.

### C.1  THE IMPROVEMENT ON CROSS-MODALITY ALIGNMENT

We first consider the loss for cross-modality alignment,

$$\mathcal{L}_{cm} = -\mathbb{E}_{(x, y_{w,o}, y_{l,o}) \sim \mathcal{D}_{cm}} \left[ \log \sigma \left( \alpha \log \frac{\pi_\theta(y_{w,o}|x)}{\pi_o(y_{w,o}|x)} - \alpha \log \frac{\pi_\theta(y_{l,o}|x)}{\pi_o(y_{l,o}|x)} \right) \right]. \tag{5}$$

where $(x_w, y_{w,o}) \sim q_w(x_w, y_{w,o}|x_t, x_r)$ and $(x_l, y_{l,o}) \sim q_l(x_l, y_{l,o}|x_t, x_r)$ represent distributions of the preferred responses and dispreferred responses on $\mathcal{D}_{cm}$, respectively. Let $x$ denote $(x_v, x_r, x_t)$

**Definition C.1** *Define the weight of $x_v$ with respect to $\log \pi_\theta(y|x)$ as*

$$wt(x_v, \pi_\theta) := \mathbb{E}_{y \sim \pi_\theta(\cdot|x)} \left[ \frac{\partial}{\partial x_v} \log \pi_\theta(y|x) \right]^2 \tag{6}$$

Definition C.1 describes how $\log \pi_\theta(y|x)$ changes with respect to $x_v$, and the weight is always non-dispreferred. We demonstrate that this is a reasonable definition through Lemma C.1.

**Lemma C.1** *For linear model $y = \theta_1 x_v + \theta_2 x_t + \epsilon$ such that $\epsilon \sim N(0, 1)$, $wt(x_v, \pi_\theta) = \theta_1^2$*

**Assumption C.1** *Let $h(x, y)$, abbreviate as $h$, be*

$$h := \left[ \sum_y \pi_o(y|x) \left( \frac{q_w(y|x)}{q_l(y|x)} \right)^{\frac{1}{\alpha}} \right]^{-1} \left( \frac{q_w(y|x)}{q_l(y|x)} \right)^{\frac{1}{\alpha}} \tag{7}$$

*Assume that $wt(x_v, \pi_o) < c^2$, where*

$$c = \sqrt{ \left\| \sqrt{\pi_o(y|x)} \cdot \frac{\partial}{\partial x_v} h \right\|_2^2 + \int \left( \frac{\partial}{\partial x_v} h \right)^2 \frac{\pi_o(y|x)}{h} dy } - \left\| \sqrt{\pi_o(y|x)} \cdot \frac{\partial}{\partial x_v} h \right\|_2 \tag{8}$$

Assumption C.1 requires that $x_v$ has a small weight in $\log \pi_o(y|x)$. A model $\pi_o(y|x)$ independent of $x_v$ could satisfy Assumption C.1. In this case, the reference model generates answers without using information from the image.

**Theorem C.1** *Suppose that Assumption C.1 holds, cross-modality loss increase the weight of $x_v$.*

$$wt(x_v, \pi_\theta) > wt(x_v, \pi_o) \tag{9}$$

Theorem C.1 indicates that when the weight of $x_v$ is too small in the initial model $\pi_o(y|x)$, the cross-modality loss function adjusts the model to place greater emphasis on images, informed by the retrieved data. Intuitively, for any sample $(x, y)$, generating unrelated images causes the policy to rely less on images. By using samples from this distribution as negative samples, the new model diverges from the initial model, increasing its reliance on images.

## C.2 The Improvement on Overall Alignment

In this section, we analyze the improvement on overall alignment. Let $q_w^1(x_v, y_{w,o}|x_t, x_r)$ and $q_l^1(x_v, y_{l,o}|x_t)$ represent distributions of the preferred responses and dispreferred responses on $\mathcal{D}_{oa}^1$, respectively; $q_w^2(x_v, y_{w,o}|x_t)$ and $q_l^2(x_v, y_{l,o}|x_t, x_r)$ represent distributions of the preferred responses and dispreferred responses on $\mathcal{D}_{oa}^2$, respectively. Overall loss is defined by

$$\mathcal{L}_{oa} = -\mathbb{E}_{(x, y_{w,o}, y_{l,o}) \sim \mathcal{D}_{oa}} \left[ \log \sigma \left( \alpha \log \frac{\pi_\theta(y_{w,o}|x)}{\pi_o(y_{w,o}|x)} - \alpha \log \frac{\pi_\theta(y_{l,o}|x)}{\pi_o(y_{l,o}|x)} \right) \right]. \tag{10}$$

Consider $\pi$ as the generative distribution underlying $\mathcal{M}$, construction of $\mathcal{D}_{oa}^1$ and $\mathcal{D}_{oa}^2$ indicate that there is a significant gap between $\pi(y|x_v, x_t, x_r)$ and $\pi(y|x_v, x_t, \tilde{x}_r)$ for $x_r$ generates true answer while $\tilde{x}_r$ generate a false one.

**Assumption C.2** *Assume that $\pi(y|x_x, x_r, x_t) : x \to y$ is L-lipschitz continuous on $x_r$ for all $(x_v, x_t, y)$ such that $|\pi(y|x_v, x_t, x_r) - \pi(y|x_v, x_t, \tilde{x}_r)| \le L \cdot d_x(x_r, \tilde{x}_r)$, where $d_x$ is any distance metric on the text space.*

Based on Assumption C.2, $\tilde{x}_r$ can be viewed as being far from the meaningful retrieved information $x_r$, resulting in different weight in the model. Then, we claim in the following theorem that the overall loss in equation 10 can effectively leverage retrieved knowledge while training.

**Assumption C.3** *Let $h_1(x_v, x_t, x_r, y)$, abbreviate as $h_1$, be*

$$h_1 := \left[ \sum_y \pi_o(y|x) \left( \frac{q_w^1(y|x_v, x_t, x_r) + q_w^2(y|x_v, x_t)}{q_l^1(y|x_v, x_t) + q_l^2(y|x_v, x_t, x_r)} \right)^{\frac{1}{\alpha}} \right]^{-1} \left( \frac{q_w^1(y|x_v, x_t, x_r) + q_w^2(y|x_v, x_t)}{q_l^1(y|x_v, x_t) + q_l^2(y|x_v, x_t, x_r)} \right)^{\frac{1}{\alpha}} \tag{11}$$

*Assume that $wt(x_r, \pi_o) < c_1^2$ and $wt(\tilde{x}_r, \pi_o) > c_2^2$, where*

$$
\begin{aligned}
c_1 &= \sqrt{\left\| \sqrt{\pi_o} \cdot \frac{\partial h_1}{\partial x_r} \right\|_2^2 + \int \left( \frac{\partial h_1}{\partial x_r} \right)^2 \frac{\pi_o}{h_1} dy} - \left\| \sqrt{\pi_o} \cdot \frac{\partial h_1}{\partial x_r} \right\|_2 \\
c_2 &= \sqrt{\left\| \sqrt{\pi_o} \cdot \frac{\partial h_1}{\partial \tilde{x}_r} \right\|_2^2 + \int \left( \frac{\partial h_1}{\partial \tilde{x}_r} \right)^2 \frac{\pi_o}{h_1} + \left( \frac{\partial \pi_o}{\partial \tilde{x}_r} \right)^2 \frac{h_1}{\pi_o} dy} + \left\| \sqrt{\pi_o} \cdot \frac{\partial h_1}{\partial \tilde{x}_r} \right\|_2
\end{aligned} \tag{12}
$$

**Theorem C.2** *Suppose that Assumption C.3 holds, then overall loss 10 increase the weight of $x_r$ and decrease the weight of $\tilde{x}_r$.*

$$wt(x_r, \pi_\theta) > wt(x_r, \pi_o), \quad wt(\tilde{x}_r, \pi_\theta) < wt(\tilde{x}_r, \pi_o) \tag{13}$$

Theorem C.2 suggests that the model tend to improve the overall alignment. When $\tilde{x}_r$ generates a false answer, the training procedure tends to reduce the reliance on $\tilde{x}_r$, resulting in a decrease in the weight assigned to $\tilde{x}_r$. Conversely, if $x_r$ is helpful for generating the true answer, $\pi_\theta(y|x)$ tend to enhance its use of $x_r$.

## D Proofs for Theoretical Results in Section C

Here we provide proofs for the results in Section C.

## D.1 Notations

Let $x_v, y, x_t, x_r$ be input medical image, ground-truth answer, question, and retrieved information, respectively. Denote $(x_w, y_{w,o}) \sim q_p(x_w, y_{w,o}|x_t, x_r)$ and $(x_l, y_{l,o}) \sim q_l(x_l, y_{l,o}|x_t, x_r)$ as distributions of the preferred responses and dispreferred responses. Let $x$ denote $(x_v, x_r, x_t)$. We aim to a fine-tune a generative model $\pi_\theta(y|x, x_t)$ through DPO loss (Rafailov et al., 2023):

$$\arg\min_{\pi_\theta} \mathbb{E}_{(x_w, x_l, y_{w,o}, y_{l,o}) \sim \mathcal{D}} U\left( \alpha \log \frac{\pi_\theta(y_{w,o}|x)}{\pi_o(y_{w,o}|x)} - \alpha \log \frac{\pi_\theta(y_{l,o}|x)}{\pi_o(y_{l,o}|x)} \right). \tag{14}$$

where $U(t) = \log(1 + \exp(-t))$. Define the weight of $x_v$ with respect to $\log \pi_\theta(y|x)$ as

$$\mathrm{wt}(x_v, \pi_\theta) := \mathbb{E}_{y \sim \pi_\theta(\cdot|x)} \left[ \frac{\partial}{\partial x_v} \log \pi_\theta(y|x) \right]^2 \tag{15}$$

## D.2 Assumptions

**Assumption D.1** *(Large parameter space) Assume that $\pi(x_v, y|x_t, x_r)$ lies in the optimization space $\{\pi_\theta, \theta \in \Theta\}$ such that $\pi(x_v, y|x_t, x_r) \propto \pi_o(x_v, y|x_t, x_r) \left( \frac{q_w(x_v, y|x_t, x_r)}{q_l(x_v, y|x_t, x_r)} \right)^{\frac{1}{\alpha}}$*

Assumption D.1 requires that the parameter space sufficiently large to ensure that $\pi_\theta$ can achieve its global optimum, allowing us to represent the optimizer with a closed form.

**Assumption D.2** *Let $h(x, y)$, abbreviate as $h$, be*

$$h := \left[ \sum_y \pi_o(y|x) \left( \frac{q_w(y|x)}{q_l(y|x)} \right)^{\frac{1}{\alpha}} \right]^{-1} \left( \frac{q_w(y|x)}{q_l(y|x)} \right)^{\frac{1}{\alpha}} \tag{16}$$

*Assume that $wt(x_v, \pi_o) < c^2$, where*

$$c = \sqrt{ \left\| \sqrt{\pi_o(y|x)} \cdot \frac{\partial}{\partial x_v} h \right\|_2^2 + \int \left( \frac{\partial}{\partial x_v} h \right)^2 \frac{\pi_o(y|x)}{h} dy } - \left\| \sqrt{\pi_o(y|x)} \cdot \frac{\partial}{\partial x_v} h \right\|_2 \tag{17}$$

**Assumption D.3** *Let $h_1(x_v, x_t, x_r, y)$, abbreviate as $h_1$, be*

$$h_1 := \left[ \sum_y \pi_o(y|x) \left( \frac{q_w^1(y|x_v, x_t, x_r) + q_w^2(y|x_v, x_t)}{q_l^1(y|x_v, x_t) + q_l^2(y|x_v, x_t, x_r)} \right)^{\frac{1}{\alpha}} \right]^{-1} \left( \frac{q_w^1(y|x_v, x_t, x_r) + q_w^2(y|x_v, x_t)}{q_l^1(y|x_v, x_t) + q_l^2(y|x_v, x_t, x_r)} \right)^{\frac{1}{\alpha}} \tag{18}$$

*Assume that $wt(x_r, \pi_o) < c_1^2$ and $wt(\tilde{x}_r, \pi_o) > c_2^2$, where*

$$c_1 = \sqrt{ \left\| \sqrt{\pi_o} \cdot \frac{\partial h_1}{\partial x_r} \right\|_2^2 + \int \left( \frac{\partial h_1}{\partial x_r} \right)^2 \frac{\pi_o}{h_1} dy } - \left\| \sqrt{\pi_o} \cdot \frac{\partial h_1}{\partial x_r} \right\|_2$$

$$c_2 = \sqrt{ \left\| \sqrt{\pi_o} \cdot \frac{\partial h_1}{\partial \tilde{x}_r} \right\|_2^2 + \int \left( \frac{\partial h_1}{\partial \tilde{x}_r} \right)^2 \frac{\pi_o}{h_1} + \left( \frac{\partial \pi_o}{\partial \tilde{x}_r} \right)^2 \frac{h_1}{\pi_o} dy } + \left\| \sqrt{\pi_o} \cdot \frac{\partial h_1}{\partial \tilde{x}_r} \right\|_2 \tag{19}$$

## D.3 Proofs

**Lemma D.1** *Suppose that Assumption D.1 hold, optimizing equation 14 gives*

$$\pi_\theta(y|x) \propto \pi_o(y|x) \left( \frac{q_w(y|x)}{q_l(y|x)} \right)^{\frac{1}{\alpha}} \tag{20}$$

Lemma D.1 indicates that the model tends to increase $\pi_o(y|x)$ if $q_w(y|x) > q_l(y|x)$, which is more likely to occur when $(x_v, y)$ represents a preferred sample given $x_t$ and $x_r$. Below, we provide an application of Lemma D.1 using a linear regression example. Lemma D.1 is proved with Lemma D.2 and Lemma D.3.

**Lemma D.2** *(Lemma C.1 in Chen et al. (2024d)) For $a, b > 0$, the following inequality holds*

$$a \cdot U(t) + b \cdot U(-t) \geq a \log(1 + b/a) + b \log(1 + a/b)$$

*and equality holds if and only if $t = \log(a/b)$*

**Lemma D.3** *Denote*

$$\begin{cases} p_1(x_w, y_{w,o}, x_l, y_{l,o}|x_t, x_r) &= q_w(x_w, y_{w,o}|x_t, x_r) \cdot q_l(x_l, y_{l,o}|x_t, x_r) \\ p_2(x_w, y_{w,o}, x_l, y_{l,o}|x_t, x_r) &= q_l(x_w, y_{w,o}|x_t, x_r) \cdot q_w(x_l, y_{l,o}|x_t, x_r) \end{cases}$$

*and abbreviated as $p_1$ and $p_2$ for notational convenience. Then,*

$$2\mathbb{E}_\mathcal{D} \left[ U \left( f(x_w, y_{w,o}, x_t, x_r) - f(x_l, y_{l,o}, x_t, x_r) \right) \right]$$

$$\geq 2 \log 2 - D_{\mathrm{KL}} \left( p_1 \| \frac{p_1 + p_2}{2} \right) - D_{\mathrm{KL}} \left( p_2 \| \frac{p_1 + p_2}{2} \right) \tag{21}$$

*Equality holds if and only if*

$$f(x, y) = g(x) + \log \frac{q_w(x_v, y|x_t, x_r)}{q_l(x_v, y|x_t, x_r)} \tag{22}$$

*where $g(x)$ is any function that is possibly dependent on $x_v$, $x_t$ and $x_r$.*

**Proof D.1**

$$2\mathbb{E}_\mathcal{D} \left[ U \left( f(x_w, y_{w,o}, x_t, x_r) - f(x_l, y_{l,o}, x_t, x_r) \right) \right]$$

$$= \int q(x_t, x_r) \cdot p_1 \cdot U \left( f(x_w, y_{w,o}, x_t, x_r) - f(x_l, y_{l,o}, x_t, x_r) \right) dxdy$$

$$+ \int q(x_t, x_r) \cdot p_2 \cdot U \left( f(x_l, y_{l,o}, x_t, x_r) - f(x_w, y_{w,o}, x_t, x_r) \right) dxdy$$

$$\geq \int q(x_t, x_r) \left[ p_1 \cdot \log \left( 1 + \frac{p_2}{p_1} \right) + p_2 \cdot \log \left( 1 + \frac{p_1}{p_2} \right) \right] dxdy \tag{23}$$

$$= 2 \log 2 + \int q(x_t, x_r) \left[ p_1 \cdot \log \left( \frac{p_1 + p_2}{2p_1} \right) + p_2 \cdot \log \left( \frac{p_1 + p_2}{2p_2} \right) \right] dxdy$$

$$= 2 \log 2 - KL \left( p_1 \| \frac{p_1 + p_2}{2} \right) - KL \left( p_2 \| \frac{p_1 + p_2}{2} \right)$$

*where the first inequality follows from Lemma D.2. For equivalence,*

$$f(x, y_{w,o}, x_t, x_r) - f(x_l, y_{l,o}, x_t, x_r) = \log \frac{q_w(x_w, y_{w,o}|x_t, x_r) \cdot q_l(x_l, y_{l,o}|x_t, x_r)}{q_l(x_w, y_{w,o}|x_t, x_r) \cdot q_w(x_l, y_{l,o}|x_t, x_r)} \tag{24}$$

*Thus, for any $x_w, y_{w,o}, x_l, y_{l,o}, x_t, x_r$,*

$$f(x_w, y_{w,o}, x_t, x_r) - \log \frac{q_w(x_w, y_{w,o}|x_t, x_r)}{q_l(x_w, y_{w,o}|x_t, x_r)} = f(x_l, y_{l,o}, x_t, x_r) - \log \frac{q_w(x_l, y_{l,o}|x_t, x_r)}{q_l(x_l, y_{l,o}|x_t, x_r)} \tag{25}$$

*Therefore, equation 25 holds if and only if there exists some $g(x_v, x_t, x_r)$ such that*

$$f(x_v, x_t, x_r, y) = g(x_t, x_r) + \log \frac{q_w(x_v, y|x_t, x_r)}{q_l(x_v, y|x_t, x_r)} \tag{26}$$

Lemma D.3 provides a closed-form solution to equation 14 if the parameter space is sufficiently large. This lemma is crucial for the proof Lemma D.1, which follows below

**Proof D.2** *According to the Assumption D.1, we have*

$$\pi(x_v, y|x_t, x_r) = \hat{g}(x_t, x_r)\pi_o(x_v, y|x_t, x_r) \left( \frac{q_w(x_v, y|x_t, x_r)}{q_l(x_v, y|x_t, x_r)} \right)^{\frac{1}{\alpha}} \tag{27}$$

*After reparameterization,*

$$\alpha \log \left( \frac{\pi(x_v, y|x_t, x_r)}{\pi_o(x_v, y|x_t, x_r)} \right) = \alpha \log[\hat{g}(x_t, x_r)] + \log \frac{q_w(x_v, y|x_t, x_r)}{q_l(x_v, y|x_t, x_r)} \tag{28}$$

*which is the global minimum of*

$$\arg\min_f \mathbb{E}_{\mathcal{D}} \left[ U \left( f(x_w, y_{w,o}, x_t, x_r) - f(x_l, y_{l,o}, x_t, x_r) \right) \right] \tag{29}$$

*by Lemma D.3. Since $\pi(x_v, y|x_t, x_r) \in \{\pi_\theta, \theta \in \Theta\}$ lies in the optimization space, we have*

$$\min_f \mathbb{E}_{\mathcal{D}} U \left( f(x_w, y_{w,o}, x_t, x_r) - f(x_l, y_{l,o}, x_t, x_r) \right)$$
$$= \min_{\pi_\theta} \mathbb{E}_{\mathcal{D}} U \left( \alpha \log \frac{\pi_\theta(y_{w,o}|x_w, x_t, x_r)}{\pi_o(y_{w,o}|x_w, x_t, x_r)} - \alpha \log \frac{\pi_\theta(y_{l,o}|x_l, x_t, x_r)}{\pi_o(y_{l,o}|x_l, x_t, x_r)} \right) \tag{30}$$

*and $\pi_\theta(x_v, y|x_t, x_r)$ is the optimizer of equation 30, which gives*

$$\alpha \log \left( \frac{\pi_\theta(x_v, y|x_t, x_r)}{\pi_o(x_v, y|x_t, x_r)} \right) = g(x_t, x_r) + \log \frac{q_w(x_v, y|x_t, x_r)}{q_l(x_v, y|x_t, x_r)}$$
$$\implies \pi_\theta(x_v, y|x_t, x_r) = \pi_o(x_v, y|x_t, x_r) \left( \frac{q_w(x_v, y|x_t, x_r)}{q_l(x_v, y|x_t, x_r)} \right)^{\frac{1}{\alpha}} \exp \left( \frac{1}{\alpha} g(x_t, x_r) \right) \tag{31}$$

*Then*

$$\pi_\theta(y|x) = \frac{\pi_\theta(x_v, y|x_t, x_r)}{\pi_\theta(x|x_t, x_r)} = \frac{\pi_o(x_v, y|x_t, x_r) \left( \frac{q_w(x_v, y|x_t, x_r)}{q_l(x_v, y|x_t, x_r)} \right)^{\frac{1}{\alpha}} \exp \left( \frac{1}{\alpha}(g(x_t, x_r)) \right)}{\sum_y \pi_o(x_v, y|x_t, x_r) \left( \frac{q_w(x_v, y|x_t, x_r)}{q_l(x_v, y|x_t, x_r)} \right)^{\frac{1}{\alpha}} \exp \left( \frac{1}{\alpha}(g(x_t, x_r)) \right)}$$
$$= \frac{\pi_o(y|x) \left( \frac{q_w(x_v, y|x_t, x_r)}{q_l(x_v, y|x_t, x_r)} \right)^{\frac{1}{\alpha}}}{\sum_y \pi_o(y|x) \left( \frac{q_w(x_v, y|x_t, x_r)}{q_l(x_v, y|x_t, x_r)} \right)^{\frac{1}{\alpha}}} = \frac{\pi_o(y|x) \left( \frac{q_w(y|x_v, x_t, x_r)}{q_l(y|x_v, x_t, x_r)} \right)^{\frac{1}{\alpha}}}{\sum_y \pi_o(y|x) \left( \frac{q_w(y|x_v, x_t, x_r)}{q_l(y|x_v, x_t, x_r)} \right)^{\frac{1}{\alpha}}} \tag{32}$$

**Corollary D.1** *Suppose that preferred responses $(x_w, y_w)$ and dispreferred responses $(x_l, y_l)$ satisfy $y_w = \beta x_w + \epsilon_1$ and $y_l = \tilde{\beta} x_l + \epsilon_2$ respectively. DPO for $y = \theta x_v + \epsilon_3$ is based on reference model $y = \theta_o x_v + \epsilon_4$, where $\epsilon_i$'s are independent and follow standard normal distribution. Then,*

$$\theta = \theta_o + \frac{1}{\alpha}(\beta - \tilde{\beta}) \tag{33}$$

Corollary D.1 is a direct application of Lemma D.1, indicating that the model updates coefficient $\theta_o$ towards the direction of $\beta$ for preferred responses and away from $\tilde{\beta}$ for dispreferred responses.

**Proof D.3** *Let $\phi(\cdot)$ denote the probability density function of standard normal, by Lemma D.1,*

$$\phi(y - \theta x) \propto \phi(y - \theta_o x) \left( \frac{\phi(y - \beta x)}{\phi(y - \tilde{\beta} x)} \right)^{\frac{1}{\alpha}}$$
$$\implies \exp \left( \frac{1}{2} y^2 - \theta_1 xy \right) \propto \exp \left( \frac{1}{2} y^2 - \theta_o xy \right) \cdot \exp \left( -\frac{1}{\alpha}(\beta - \tilde{\beta}) xy \right)$$
$$\implies \exp(\theta_1 xy) \propto \exp(\theta_o xy) \cdot \exp \left( \frac{1}{\alpha}(\beta - \tilde{\beta}) xy \right) \tag{34}$$
$$\implies \theta = \theta_o + \frac{1}{\alpha}(\beta - \tilde{\beta})$$

**Lemma D.4** *For linear model $y = \theta_1 x_v + \theta_2 x_t + \epsilon$ such that $\epsilon \sim N(0, 1)$, $wt(x_v, \pi_\theta) = \theta_1^2$*

**Proof D.4** *Let $\phi(\cdot)$ denote the probability density function of standard normal,*

$$
\begin{aligned}
wt(x_v, \pi_\theta) &= \int \left( -\frac{1}{2} \frac{\partial}{\partial x_v} (y - \theta_1 x_v - \theta_2 x_t)^2 \right)^2 \phi(y - \theta_1 x_v - \theta_2 x_t) dy \\
&= \theta_1^2 \int (y - \theta_1 x_v - \theta_2 x_t)^2 \phi(y - \theta_1 x_v - \theta_2 x_t) dy \\
&= \theta_1^2 \int (\theta_1 x_v + \theta_2 x_t - y) \frac{d\phi(y - \theta_1 x_v - \theta_2 x_t)}{dy} dy \\
&= \theta_1^2 \int \phi(y - \theta_1 x_v - \theta_2 x_t) dy = \theta_1^2
\end{aligned}
\tag{35}
$$

**Theorem D.2** *Suppose that Assumption D.2 holds, then cross-modality increase the weight of $x_v$.*

$$
wt(x_v, \pi_\theta) > wt(x_v, \pi_o)
\tag{36}
$$

**Proof D.5** *By Lemma D.1, we have*

$$
\pi_\theta(y|x) = \pi_o(y|x) \cdot h(x, y), \quad \int \pi_o(y|x) \cdot h(x,y) dy = 1
\tag{37}
$$

*Abbreviate $h(x,y)$ and $\pi_o(y|x_v, x_t)$ as $h$ and $\pi_o$ respectively, we have*

$$
\begin{aligned}
wt(x_v, \pi_\theta) - wt(x_v, \pi_o) &\geq \int \left( \frac{\frac{\partial}{\partial x_v} \pi_o}{\pi_o} + \frac{\frac{\partial}{\partial x_v} h}{h} \right)^2 \pi_o h \, dy - wt(x_v, \pi_o) \\
&\geq \int \left[ \frac{\partial}{\partial x_v} h \right]^2 \frac{\pi_o}{h} dy - 2\sqrt{wt(x_v, \pi_o)} \cdot \left\| \sqrt{\pi_o} \cdot \frac{\partial}{\partial x_v} h \right\|_2 - wt(x_v, \pi_o)
\end{aligned}
\tag{38}
$$

*the second inequality follows from Cauchy–Schwarz inequality*

$$
\int \frac{\partial}{\partial x_v} \pi_o \cdot \frac{\partial}{\partial x_v} h \, dy = \int \frac{\partial}{\partial x_v} \pi_o \cdot \frac{\sqrt{\pi_o}}{\sqrt{\pi_o}} \cdot \frac{\partial}{\partial x_v} h \, dy \leq \sqrt{wt(x_v, \pi_o)} \cdot \left\| \sqrt{\pi_o} \cdot \frac{\partial}{\partial x_v} h \right\|_2
\tag{39}
$$

*Denote $c$ as*

$$
c := \sqrt{ \left\| \sqrt{\pi_o} \cdot \frac{\partial}{\partial x_v} h \right\|_2^2 + \int \left( \frac{\partial}{\partial x_v} h \right)^2 \frac{\pi_o}{h} dy } - \left\| \sqrt{\pi_o} \cdot \frac{\partial}{\partial x_v} h \right\|_2
\tag{40}
$$

*the last term in equation 38 is equivalent to*

$$
\left( c - \sqrt{wt(x_v, \pi_o)} \right) \cdot \left( \sqrt{wt(x_v, \pi_o)} + c + 2 \left\| \sqrt{\pi_o} \cdot \frac{\partial}{\partial x_v} h \right\|_2 \right)
\tag{41}
$$

*Thus, $wt(x_v, \pi_\theta) > wt(x_v, \pi_o)$ if $\sqrt{wt(x_v, \pi_o)} < c$.*

**Theorem D.3** *Suppose that Assumption D.3 holds, the overall loss increase the weight of $x_r$ and decrease the weight of $\tilde{x}_r$.*

$$
wt(x_r, \pi_\theta) > wt(x_r, \pi_o), \quad wt(\tilde{x}_r, \pi_\theta) < wt(\tilde{x}_r, \pi_o)
\tag{42}
$$

**Proof D.6** *The distribution of preferred responses can be considered as a mixture distribution: $q_w^1(x_v, y_{w,o}|x_t, x_r) + q_w^2(x_v, y_{w,o}|x_t)$. Similarly, for dispreferred responses, the distribution is represented as $q_l^1(x_v, y_{l,o}|x_t) + q_l^2(x_v, y_{l,o}|x_t, x_r)$. By Lemma D.1,*

$$
\pi_\theta(y|x) = \pi_o(y|x) \cdot h_1(x, y), \quad \int \pi_o(y|x) \cdot h_1(x,y) dy = 1
\tag{43}
$$

*Abbreviate $h_1(x,y)$ as $h_1$. Follow the same procedure in the proof of Theorem D.2,*

$$
\begin{aligned}
wt(x_r, \pi_\theta) - wt(x_r, \pi_o) &\geq \int \left[ \frac{\partial}{\partial x_r} h_1 \right]^2 \frac{\pi_o}{h_1} dy - 2\sqrt{wt(x_r, \pi_o)} \cdot \left\| \sqrt{\pi_o} \cdot \frac{\partial}{\partial x_r} h_1 \right\|_2 - wt(x_r, \pi_o) \\
&= \left( c_1 - \sqrt{wt(x_r, \pi_o)} \right) \cdot \left( \sqrt{wt(x_r, \pi_o)} + c_1 + 2 \left\| \sqrt{\pi_o} \cdot \frac{\partial}{\partial x_r} h_1 \right\|_2 \right)
\end{aligned}
\tag{44}
$$

*where we apply Cauchy–Schwarz inequality in equation 44.*

$$c_1 = \sqrt{\left\| \sqrt{\pi_o(y|x)} \cdot \frac{\partial}{\partial x_r} h_1 \right\|_2^2 + \int \left( \frac{\partial}{\partial x_r} h_1 \right)^2 \frac{\pi_o(y|x)}{h_1} dy} - \left\| \sqrt{\pi_o(y|x)} \cdot \frac{\partial}{\partial x_r} h_1 \right\|_2 \quad (45)$$

*Thus, $wt(x_r, \pi_\theta) > wt(x_r, \pi_o)$ if $\sqrt{wt(x_r, \pi_o)} < c_1$. Again, by Cauchy–Schwarz inequality*

$$wt(\tilde{x}_r, \pi_\theta) - wt(\tilde{x}_r, \pi_o)$$

$$\leq \int \left( \frac{\partial h_1}{\partial \tilde{x}_r} \right)^2 \frac{\pi_o}{h_1} + \left( \frac{\partial \pi_o}{\partial \tilde{x}_r} \right)^2 \frac{h_1}{\pi_o} dy + 2\sqrt{wt(\tilde{x}_r, \pi_o)} \cdot \left\| \sqrt{\pi_o} \cdot \frac{\partial h_1}{\partial \tilde{x}_r} \right\|_2 - wt(\tilde{x}_r, \pi_o) \quad (46)$$

$$= -\left( \sqrt{wt(\tilde{x}_r, \pi_o)} - c_2 \right) \cdot \left( \sqrt{wt(\tilde{x}_r, \pi_o)} - c_2 + 2 \left\| \sqrt{\pi_o} \cdot \frac{\partial}{\partial \tilde{x}_r} h_1 \right\|_2 \right)$$

*where*

$$c_2 = \sqrt{\left\| \sqrt{\pi_o} \cdot \frac{\partial}{\partial \tilde{x}_r} h_1 \right\|_2^2 + \int \left( \frac{\partial}{\partial \tilde{x}_r} h_1 \right)^2 \frac{\pi_o}{h_1} + \left( \frac{\partial}{\partial \tilde{x}_r} \pi_o \right)^2 \frac{h_1}{\pi_o} dy} + \left\| \sqrt{\pi_o} \cdot \frac{\partial}{\partial \tilde{x}_r} h_1 \right\|_2 \quad (47)$$

*Thus, $wt(x_r, \pi_\theta) < wt(x_r, \pi_o)$ if $\sqrt{wt(x_r, \pi_o)} > c_2$.*

