# OpenReview forum: "MMed-RAG: Versatile Multimodal RAG System for Medical Vision Language Models"
_NeurIPS.cc/2024/Workshop/SafeGenAi — SafeGenAi Poster_

### Official Review · Reviewer_qczy · 2024-10-09
**The paper introduces MMed-RAG a method that augments RAG in MLVLMs by modifying the retrieval mechanism and introducing preference based fine-tuning to increase factuality. Great paper with elegant solutions to interesting problems regarding hallucination in MLVLMs.**

**Rating:** 8
**Confidence:** 4

**Review:**

This is a high-quality paper with a comprehensive experimental design to evaluate the ability of their approach to produce, factual medical evaluations in comparison to existing methods. The paper is written very clearly, making the experiments and methods easy to understand and potentially replicate.

Pros:
- The paper demonstrates that they significantly surpass the factuality of existing methods (43.8%) when utilising their modified RAG framework.
- The approach is versatile and can be used across many medical domains (ophthalmology, radiology, pathology).
- The paper presents elegant solutions to several problems: high-quality retrieval, number of retrieved examples, cross-modality alignment and overall alignment.
- The provability of the RAG based preference fine-tuning is a nice touch.

Cons:
- It would be nice to see further exploration of the thresholding parameter, \gamma for adaptive context selection and also the effect of using different noising schedules in the cross modality alignment section.
- The performance of MMed-RAG relies heavily on high-quality labeled datasets for the retrieval mechanism and fine-tuning, which can be a limitation in medical fields where such datasets are not always available.
- Some small points on grammar/clarity:
Line 086 should be modified to 'utilizes preference data to achieve'.
Is there a reason why the Similarity matrix in the exponent of the denominator of L_{txt}, has switched indices in comparison to L_{img}? If not its clear to keep them consistent.
Line 150 should be modified to 'multimodal retriever retrieves'.

---

### Official Review · Reviewer_AQDL · 2024-10-09
**Multimodal RAG system enhancing factuality of Med-LVLMs by 43.8%**

**Rating:** 8
**Confidence:** 3

**Review:**

The authors proposed a multimodal RAG system that is generalizable to different specialty and alleviates the cross-modality and general misalignment issues. The paper is very well written with good problem statement, prior work survey, system details and experiments. The experiments covers multiple medical dataset and compares the result with other Med-LVLMs, which is very convincing. It would be nice to see different alternative for each module in the system, if any. Also it would be great to mention the trade off for such a system and also the future work.

---

### Official Review · Reviewer_rorA · 2024-10-10
**Review of the MMed-RAG paper**

**Rating:** 7
**Confidence:** 3

**Review:**

This paper explores different strategies for implementing RAG in a vision-language model for medical applications. The authors propose three new strategies for enhancing RAG-based model's performance: domain identifier, adaptive top-k context retrieval, and preference finetuning. The manuscript provides comprehensive evaluations across multiple datasets and baselines, ablation studies, and includes mathematical justifications in the appendix. Overall, I find this paper interesting and well-written, and I recommend its acceptance with a few minor suggestions listed below.

1. The manuscript does not explicitly discuss known bugs and limitations.

2. I would like to see a more general statement about the applicability of the proposed method to other RAG-based models. For example, the authors’ speculation on adaptability in other domains and potential issues that might arise.

3. The authors compared various baselines, some of which are based on RAG. I wonder if the preference finetuning proposed in this study could be applied to other RAG-based baseline models and further enhance their performance, especially for RULE, which performs closely to the authors' method in Tables 1 and 2.

4.	There are some technical details that are either missing or unclear:
- (Eq. 1) \pi_{ref} is not defined.
- (Eq. 2) In this SimCLR-style loss, it’s unclear whether the exclusion of an indicator function for i==j was intentional or a mistake.
- (Eq. 4) \pi_o is not defined. As mentioned earlier for Eq. 1, I assume this refers to a pre-finetuned model, but an explicit mention is required for clarity.
- (Fig. 3) The "Copy-Reference" and "Over-Reliance" are not quantitatively defined in the caption or in the text.
- (Fig. 4) Since this is just one sample, it is insufficient to justify the authors’ claim. I suggest using a metric that can capture general behavior across the dataset. For example, presenting changes in the mean value of attention scores (after normalization) for image tokens versus text tokens could strengthen the argument.

---

### Official Review · Reviewer_C3w5 · 2024-10-12
**This paper proposes a novel multimodal RAG system for Med-LVLMs, demonstrating significant improvements in factual accuracy**

**Rating:** 8
**Confidence:** 4

**Review:**

**Summary**:
This paper presents a novel multimodal RAG system for Med-LVLMs. The system features domain-aware retrieval, adaptive selection of retrieved contexts, and preference-based fine-tuning. Extensive experiments demonstrate substantial improvements in the factual accuracy of Med-LVLMs across different medical imaging domains.

**Strengths**:
1. The proposed multimodal RAG system is innovative, especially in its use of preference pairs to address cross-modality misalignment and improve robustness against noisy inputs. This approach also offers potential for self-improvement in Med-LVLMs.
2. The method is strongly supported by comprehensive experiments, showing significant improvements. The ablation study effectively validates the contribution of each component in the framework.
3. The writing is clear, well-organized, and easy to follow.

**Weakness**:
1. Including experiments comparing this method with standard RAG fine-tuning on the same dataset would provide a more thorough evaluation.